# Outcome after Surgery for Iatrogenic Acute Type A Aortic Dissection

**DOI:** 10.3390/jcm11226729

**Published:** 2022-11-14

**Authors:** Fausto Biancari, Matteo Pettinari, Giovanni Mariscalco, Caius Mustonen, Francesco Nappi, Joscha Buech, Christian Hagl, Antonio Fiore, Joseph Touma, Angelo M. Dell’Aquila, Konrad Wisniewski, Andreas Rukosujew, Andrea Perrotti, Amélie Hervé, Till Demal, Lenard Conradi, Marek Pol, Petr Kacer, Francesco Onorati, Cecilia Rossetti, Igor Vendramin, Daniela Piani, Mauro Rinaldi, Luisa Ferrante, Eduard Quintana, Robert Pruna-Guillen, Javier Rodriguez Lega, Angel G. Pinto, Timo Mäkikallio, Metesh Acharya, Zein El-Dean, Mark Field, Amer Harky, Sebastien Gerelli, Dario Di Perna, Mikko Jormalainen, Giuseppe Gatti, Enzo Mazzaro, Tatu Juvonen, Sven Peterss

**Affiliations:** 1Department of Cardiac Surgery, Heart and Lung Center, Helsinki University Hospital, University of Helsinki, 00029 Helsinki, Finland; 2Department of Medicine, South-Karelia Central Hospital, University of Helsinki, 53130 Lappeenranta, Finland; 3Department of Cardiac Surgery, Ziekenhuis Oost Limburg, 3600 Genk, Belgium; 4Department of Cardiac Surgery, Glenfield Hospital, Leicester LE3 9QP, UK; 5Department of Cardiac Surgery, Centre Cardiologique du Nord de Saint-Denis, 93200 Paris, France; 6Department of Cardiac Surgery, LMU University Hospital, Ludwig Maximilian University, 80539 Munich, Germany; 7German Centre for Cardiovascular Research, Partner Site Munich Heart Alliance, 80539 Munich, Germany; 8Department of Cardiac Surgery, Hôpitaux Universitaires Henri Mondor, Assistance Publique-Hôpitaux de Paris, 94000 Creteil, France; 9Department of Vascular Surgery, Hôpitaux Universitaires Henri Mondor, Assistance Publique-Hôpitaux de Paris, 94000 Creteil, France; 10Department of Cardiothoracic Surgery, University Hospital Muenster, 48149 Muenster, Germany; 11Department of Thoracic and Cardiovascular Surgery, University of Franche-Comte, 25030 Besancon, France; 12Department of Cardiovascular Surgery, University Heart & Vascular Center Hamburg, 20251 Hamburg, Germany; 13Department of Cardiac Surgery, Third Faculty of Medicine, Charles University and University Hospital Kralovske Vinohrady, 10000 Prague, Czech Republic; 14Division of Cardiac Surgery, Medical School, University of Verona, 37124 Verona, Italy; 15Cardiothoracic Department, University Hospital of Udine, 33100 Udine, Italy; 16Cardiac Surgery, Molinette Hospital, University of Turin, 10126 Turin, Italy; 17Department of Cardiovascular Surgery, Hospital Clínic de Barcelona, University of Barcelona, 08036 Barcelona, Spain; 18Cardiovascular Surgery Department, University Hospital Gregorio Marañón, 28007 Madrid, Spain; 19Liverpool Centre for Cardiovascular Sciences, Liverpool Heart and Chest Hospital, Liverpool L14 3PE, UK; 20Centre Hospitalier Annecy Genevois, 74370 Annecy, France; 21Division of Cardiac Surgery, Cardio-Thoracic and Vascular Department, Azienda Sanitaria Universitaria Giuliano Isontina, 34148 Trieste, Italy; 22Research Unit of Surgery, Anesthesia and Critical Care, University of Oulu, 90570 Oulu, Finland

**Keywords:** type A aortic dissection, aortic dissection, iatrogenic, emergency

## Abstract

(1) Background: Acute Stanford type A aortic dissection (TAAD) may complicate the outcome of cardiovascular procedures. Data on the outcome after surgery for iatrogenic acute TAAD is scarce. (2) Methods: The European Registry of Type A Aortic Dissection (ERTAAD) is a multicenter, retrospective study including patients who underwent surgery for acute TAAD at 18 hospitals from eight European countries. The primary outcomes were in-hospital mortality and 5-year mortality. Twenty-seven secondary outcomes were evaluated. (3) Results: Out of 3902 consecutive patients who underwent surgery for acute TAAD, 103 (2.6%) had iatrogenic TAAD. Cardiac surgery (37.8%) and percutaneous coronary intervention (36.9%) were the most frequent causes leading to iatrogenic TAAD, followed by diagnostic coronary angiography (13.6%), transcatheter aortic valve replacement (10.7%) and peripheral endovascular procedure (1.0%). In hospital mortality was 20.5% after cardiac surgery, 31.6% after percutaneous coronary intervention, 42.9% after diagnostic coronary angiography, 45.5% after transcatheter aortic valve replacement and nihil after peripheral endovascular procedure (*p* = 0.092), with similar 5-year mortality between different subgroups of iatrogenic TAAD (*p* = 0.710). Among 102 propensity score matched pairs, in-hospital mortality was significantly higher among patients with iatrogenic TAAD (30.4% vs. 15.7%, *p* = 0.013) compared to those with spontaneous TAAD. This finding was likely related to higher risk of postoperative heart failure (35.3% vs. 10.8%, *p* < 0.0001) among iatrogenic TAAD patients. Five-year mortality was comparable between patients with iatrogenic and spontaneous TAAD (46.2% vs. 39.4%, *p* = 0.163). (4) Conclusions: Iatrogenic origin of acute TAAD is quite uncommon but carries a significantly increased risk of in-hospital mortality compared to spontaneous TAAD.

## 1. Introduction

Acute Stanford type A aortic dissection (TAAD) is a severe emergency condition, which is associated with significant mortality and morbidity [1,2]. Surgery for TAAD is often performed as a salvage procedure. TAAD is mostly of spontaneous origin, but it may develop as a result of intimal injury during cardiac surgery, interventional cardiology procedures or diagnostic coronary angiography [3]. The incidence of acute TAAD during percutaneous coronary intervention or diagnostic coronary angiography has been observed ranging from 0.02% to 0.06% [4,5,6], while its incidence is higher after cardiac surgery, as it ranges from 0.06% to 0.2% [7,8]. Ram et al. [2] estimated that the incidence of iatrogenic TAAD (iTAAD) may be 0.47% to 0.49% after cardiac surgery with femoral or subclavian arterial cannulation, respectively. The International Registry of Aortic Dissection (IRAD) investigators reported a prevalence of iatrogenic aortic dissection of 5%. TAAD represented 76% of iatrogenic aortic dissections in the registry and these were secondary to cardiac surgery in 69% of cases [9]. In the IRAD registry, early mortality after surgery for iTAAD was comparable to that of spontaneous TAAD (sTAAD) (32% vs. 35%, respectively) [9]. More recently, the German Registry for Acute Aortic Dissection Type A (GERAADA) reported on a prevalence of iTAAD of 4.7% with 30-day mortality comparable to sTAAD (16% vs. 17%, respectively) [10]. However, these studies did not evaluate the outcome of iTAAD patients after the recent widespread use of transcatheter procedures [11] and peripheral cannulation for minimally invasive cardiac surgery [7]. In this study, we aimed to evaluate the outcomes of patients who required surgery for acute TAAD secondary to iatrogenic injury from a multicenter, observational study.

## 2. Materials and Methods

### 2.1. Study Design

The European Registry of Type A Aortic Dissection (ERTAAD) is an observational, multicenter, retrospective cohort study, which was approved by the Ethical Review Board of the Helsinki University Hospital, Finland (21 April 2021, diary no. HUS/237/2021), and by the Ethical Review Board of each participating hospital. The requirement for informed consent was waived because of the retrospective nature of this study. The ERTAAD registry included consecutive patients who underwent surgery for acute TAAD at 18 cardiac surgery centers in eight European countries (Belgium, Czech Republic, Finland, France, Germany, Italy, Spain and the United Kingdom) from 1 January 2005 to 31 March 2021. Data was retrospectively collected into a Microsoft Access datasheet (Redmond, Washington, DC, USA) with pre-specified baseline, operative and outcome variables. Data on the date of death were collected retrospectively from electronic national registries, as well as by contacting regional hospitals, patients and their relatives.

### 2.2. Participants

The study participants were recruited according to the following inclusion criteria: (1) TAAD involving the ascending aorta; (2) patients aged  >18 years; (3) symptoms started within 7 days of surgery; (4) primary surgical repair of acute TAAD; (5) any other major cardiac surgical procedure concomitant with surgery for TAAD [12]. The exclusion criteria were the following: (1) patients aged <18 years; (2) onset of symptoms > 7 days from surgery; (3) prior procedure for TAAD; (4) retrograde TAAD (with primary tear located in the descending aorta); (5) concomitant endocarditis; (6) TAAD secondary to blunt or penetrating chest trauma [12]. The diagnosis of acute iTAAD was made based on clinical conditions, periprocedural imaging and intraoperative findings.

### 2.3. Risk Factor Criteria

The definition criteria for risk factors of interest have been previously reported [12]. We defined malperfusion as acute organ ischemia secondary to aortic branch vessel hypoperfusion. This severe condition is usually classified based on clinical signs and symptoms related to ischemia [13,14]. Cerebral malperfusion was defined as unconsciousness before sedation, hemiplegia/hemiparesis, dysarthria/aphasia, vision loss, gaze deviation and confusion which was believed being related to cerebral ischemia. Spinal malperfusion was defined as acute paraparesis/paraplegia. Mesenteric malperfusion was defined as a sudden, mild-to-severe abdominal pain with or without nausea and vomiting, which was accompanied or not by rectal bleeding or bloody diarrhea. Renal malperfusion was defined as anuria/oliguria. Peripheral malperfusion was defined as loss of pulse with or without sensory or motor deficits of any limb. Salvage procedure was defined as any surgical operation performed on patients with cardiogenic shock requiring cardiac massage en route to the operating room or after anesthesia induction [12].

### 2.4. Interventions

Aortic root replacement referred to procedures which included resection of the aortic root, with or without preservation of the aortic valve, i.e., the Bentall–De Bono procedure, the David procedure and the Yacoub procedure. Interventions not involving resection of the aortic root, including also the partial Yacoub procedure and the Florida sleeve procedure, were defined as supracoronary replacement procedures. Aortic arch replacement procedures were those implying partial or total resection of the aortic arch with distal anastomosis to the Ishimaru zones 1 to 4 with reimplantation of at least one aortic arch vessel originating from the aortic arch.

### 2.5. Outcome Measures

Primary outcomes of this study are mortality during index hospitalization and at 5 years. Secondary outcomes were stroke, global brain ischemia, paraplegia/paraparesis and tetraplegia/tetraparesis, as well as a composite endpoint including in-hospital mortality, stroke and global brain ischemia. Other secondary outcomes were heart failure, need of mechanical circulatory support, use of intra-aortic balloon pump, use of venoarterial extracorporeal membrane oxygenation, dialysis, sepsis, laryngeal nerve palsy, reoperation for intrathoracic bleeding, tracheostomy, deep sternal wound infection, mesenteric ischemia, limb ischemia and procedures for vascular and intestinal complications. Definition criteria of these outcomes have been previously reported in detail [12].

### 2.6. Statistical Analysis

Continuous variables are reported as means and standard deviations and nominal variables as counts and percentages. The chi-square test and Fisher’s exact test were used to analyze differences between categorical variables, and the Mann–Whitney test and Kruskall–Wallis test to compare continuous variables. Survival analysis was performed using the Kaplan–Meier test and Cox proportional hazards method. Analysis of aortic reoperations was performed using competing risk analysis with the Fine–Gray test and all-cause death as a competing event. Risk estimates are reported as hazard ratios (HR) or subdistributional hazard ratios (SHRs) and 95% confidence interval (CI). Baseline characteristics differed significantly among patients with and without iTAAD. Therefore, a propensity score matching analysis was used to adjust for imbalances between the study groups. A propensity score was estimated using logistic regression with the iTAAD vs. sTAAD as the dependent variable considering the following covariates: age, gender, bicuspid aortic valve, diabetes, stroke, pulmonary disease, extracardiac arteriopathy, prior cardiac surgery, preoperative cardiac massage, cardiogenic shock requiring inotropes, cerebral malperfusion, spinal malperfusion, renal malperfusion, mesenteric malperfusion, peripheral malperfusion, DeBakey I aortic dissection, salvage procedure, aortic root replacement and aortic arch replacement. Coronary artery bypass grafting was not included into the logistic regression model because most of the procedures resulting in TAAD were performed on coronary arteries, and their injury might have accompanied dissection of the aorta. Propensity score matching was performed using the psmatch2 module for Stata with a caliper width of 0.2 the standard deviation of the logit. Standardized difference <0.1 was considered a non-significant imbalance between the covariates. We evaluated the results of earlier and more recent procedures by comparing the results of patients operated on before 2014 and those operated on later, adjusted for independent risk factors for in-hospital mortality, i.e., participating hospitals, age, preoperative cardiac massage, cerebral malperfusion, mesenteric malperfusion, peripheral malperfusion, aortic root replacement and partial/total aortic arch replacement. Statistical significance was set at a level of *p* < 0.05. Statistical analyses were performed with the SPSS (version 27.0, SPSS Inc., IBM, Chicago, Illinois, USA) and the Stata (version 15.1, StataCorp LLC, College Station, TX, USA) statistical software.

## 3. Results

### 3.1. Patient Population

Overall, 3902 consecutive patients who underwent surgery for acute TAAD were included in the present study. The mean age of patients was 63.3 ± 13.0 years, and 1185 (30.4%) of patients were females. iTAAD was observed in 103 (2.6%) patients and their characteristics are summarized in Table 1. Patients with iTAAD were significantly older (mean age 69.2 vs. 63.2 years, *p* < 0.0001) and more frequently female (45.6% vs. 30.0%, *p* = 0.001) than patients with sTAAD. DeBakey type I aortic dissection was less frequent among iTAAD (68.0% vs. 84.4%, *p* < 0.0001). Patients with iTAAD and sTAAD significantly differed in several baseline and operative covariates. Cardiac surgery (39 patients, 37.8%) and percutaneous coronary intervention (38 patients, 36.9%) were the most frequent causes leading to iTAAD, followed by diagnostic coronary angiography (14 patients, 13.6%), transcatheter aortic valve replacement (11 patients, 10.7%) and peripheral endovascular procedure (1 patient, 1.0%) (Table 2 and Table 3). The delay from onset of symptoms to surgery for iTAAD was estimated to be 10.5 ± 19.2 h (median 3.0 h, range 0–96 h), and from the primary procedure to surgery for iTAAD was 1.4 ± 4.4 days (median 0 days, range 0–36 days). Details on patients’ characteristics and operative data of patients according to the cause of TAAD are reported in Table 2. Patients’ age significantly differed between different procedures resulting in iTAAD as patients who underwent cardiac surgery and transcatheter aortic valve replacement were significantly older than patients in the other subgroups. Coronary artery bypass grafting and valve surgery was equally prevalent among the causes leading to TAAD.

Among patients with iTAAD, aortic root replacement was performed in 27.3% of patients after transcatheter aortic valve replacement and 28.6% of patients after diagnostic coronary angiography, while it was less frequent in the other subgroups of iTAAD (Table 2). Coronary artery bypass grafting was required in 63.2% of patients after percutaneous coronary intervention, in 35.7% of patients after diagnostic coronary angiography and 43.6% of cardiac surgery patients. Coronary artery bypass grafting was not needed after transcatheter aortic valve replacement (Table 2).

### 3.2. Outcomes in the Overall Series

Crude in-hospital mortality (30.1% vs. 17.3%, *p* = 0.001) and 5-year mortality (45.6% vs. 32.2%, *p* = 0.002) were significantly higher in patients who underwent surgery for iTAAD compared to sTAAD. Five-year mortality was similar between patients with different causes of iTAAD (*p* = 0.710, Figure 1). Patients with iTAAD had an increased risk of postoperative heart failure and of mechanical circulatory support, but no difference was observed in other secondary outcomes compared to sTAAD patients (Table 4). In-hospital mortality was 20.5% after cardiac surgery, 31.6% after percutaneous coronary intervention, 42.9% after diagnostic coronary angiography, 45.5% after transcatheter aortic valve replacement and nihil after peripheral endovascular procedure (*p* = 0.092) (Table 3), with similar 5-year mortality between patients with different causes of iTAAD (*p* = 0.710) (Figure 1).

### 3.3. Outcomes of Propensity Score Matched Pairs

Propensity score matching with a caliper width of 0.22 yielded 102 pairs of patients with spontaneous and iTAAD with balanced baseline and operative covariates (Table 1). In-hospital mortality was significantly higher among patients with iTAAD (30.4% vs. 15.7%, *p* = 0.013) compared to those with sTAAD. The increased early mortality of patients with iTAAD was most likely due to their higher risk of postoperative heart failure (35.3% vs. 10.8%, *p* < 0.0001), which required more frequently the use of mechanical circulatory support (12.7% vs. 2.0%, *p* = 0.005). No difference was observed between the study cohorts in the other early outcomes (Table 4). At 5-year, patients with iTAAD had numerically higher mortality rates (46.2% vs. 39.4%, *p* = 0.163), but the difference did not reach statistical significance.

### 3.4. Earlier vs. More Recent Procedures

We evaluated the results of earlier and more recent procedures by comparing the results of patients operated before 2014 (44 patients, 42.7%) and those operated on later (59 patients, 57.3%), adjusted for independent risk factors for in-hospital mortality along with types of procedures. Patients operated on more recently did not have a lower risk of in-hospital mortality (*p* = 0.161, OR 3.330, 95%CI 0.619–17.901).

## 4. Discussion

The results of the present study can be summarized as follows: (1) the prevalence of iTAAD in this large multicenter study was rather low; (2) iTAAD is associated with a significantly increased risk of early mortality; (3) transcatheter aortic valve replacement is an emerging cause of iTAAD and it is associated with excessive early mortality.

The prevalence of iTAAD in the ERTAAD registry was 2.6%, which is lower than that observed in the GERAADA registry (4.7%) [10]. The IRAD registry reported on a prevalence of iTAAD of 3.6%, but it is unclear whether all these patients underwent surgery [9]. We speculate that nowadays surgery is offered mainly to patients with extensive aortic dissection and/or symptoms. Clinical series and several case reports demonstrated that iTAAD of limited extent can be treated conservatively with good results [4,5,15,16,17,18]. Regarding the value of conservative treatment strategy, a pooled analysis by Jonker et al. [19] showed that when iTAAD was not surgically treated, mortality was 86% compared to 35% after surgery. Indeed, surgery remains the main treatment method for those patients with extensive iTAAD [20]. Since negative results are expected to be less frequently reported, more conclusive results are needed to define the types of TAAD which may be safely treated conservatively.

Previous reports from the IRAD registry [9] and the GERAADA registry [10] reported comparable early mortality after surgery for iTAAD and sTAAD. However, the present study showed that either unadjusted or adjusted in-hospital mortality was significantly higher in patients with iTAAD. Interestingly, patients with iTAAD suffered more frequently of postoperative heart failure requiring mechanical circulatory support (Table 3). This finding is clinically sound, as we might expect that either dissection of coronary arteries during percutaneous coronary procedures, or prolonged myocardial ischemia for repair of aortic dissection, along with other primary heart diseases, might have resulted in significant myocardial ischemic injury, which translated into increased mortality rate.

The present series showed that early mortality was rather low among patients who underwent repair of iTAAD after cardiac surgery (in-hospital mortality: 20.5%; 30-day mortality: 23.1%). This result compared well with a larger series, which reported early mortality up to 41.5% (6). Such a difference in mortality persisted along the years because 5-year mortality was 47.5% in our series (Figure 1), while it was 60% in Leontyev’s study [6]. A multicenter study by Stanger et al. [21] reported a mortality of 27.8% early after surgery for post-cardiac surgery iTAAD with a 5-year mortality of 41%.

Previous studies did not include iTAAD secondary to transcatheter aortic valve replacement, which certainly contributed to the high early mortality rate of this series. Indeed, transcatheter aortic valve replacement was associated with 45.5% in-hospital mortality. A pooled analysis showed a risk of aortic dissection/aortic injury of 0.15% [22]. In the study by Eggebrecht et al. [11], the risk of iatrogenic aortic dissection was about 0.1%, without decrease after the introduction of newer generation transcatheter aortic valve devices. Emergency surgery for aortic dissection was associated with in-hospital mortality of 52%, which is comparable to that of the present study (45.5%). Transcatheter aortic valve replacement is usually performed in intermediate- or high-risk patients with advanced age, and this may explain the excessive mortality rate of these patients after surgery for iTAAD.

Data on aortic diseases underlying the development of iTAAD was not available in this study. Since iTAAD may be a subject of medico-legal litigation, any diseases of the aorta may be of relevance in discussing the pathogenesis of TAAD secondary to maneuvers on the aorta during cardiac surgery or endovascular procedures. Stanger et al. [21] reported that only 7.4% of aortic tissue was normal at histopathologic examination, while 44.4% had medial degeneration, 33.3% atherosclerosis, 11.1% a combination of atherosclerosis and medial degeneration and 3.7% had giant-cell arteritis. Leontyev et al. [6] reported on atherosclerotic changes in 61.1% of patients, cystic medial necrosis in 22.2% and aortitis in 2.8% of patients operated for iTAAD. These findings suggest that iTAAD is most often related to diseases of the aorta and mechanical injury is a trigger of such a severe complication.

The retrospective nature is the main limitation of this study. Second, the outcomes might have been affected by the individual surgeon’s experience with aortic surgery, interinstitutional differences in the treatment of these critically ill patients and referral pathways. However, the limited number of patients with iTAAD prevented an analysis of these factors on the results. Third, the ERTAAD registry did not include data on the causes and sites causing TAAD. Fourth, the decision on whether to surgically repair iTAAD was based on patient’s clinical conditions and the extent of aortic dissection at imaging. Therefore, we may expect that this study captured only patients with extensive TAAD, who could have not been treated conservatively. Finally, despite the relatively large size of this registry and the rather high odds ratio (2.347) of hospital mortality between the matched groups, posthoc power analysis showed that the number of iatrogenic TAAD might not be enough to reject the null hypothesis. In fact, the estimated sample size for a matched case-control study should be 126 cases per study group (alpha: 0.050, Beta: 0.80).

## 5. Conclusions

In conclusion, this multicenter study showed that iTAAD requiring surgical repair is a rather uncommon condition, but it carries an increased risk of early mortality. Transcatheter aortic valve replacement is an emerging cause of iTAAD and may be associated with high early mortality.

## Figures and Tables

**Figure 1 jcm-11-06729-f001:**
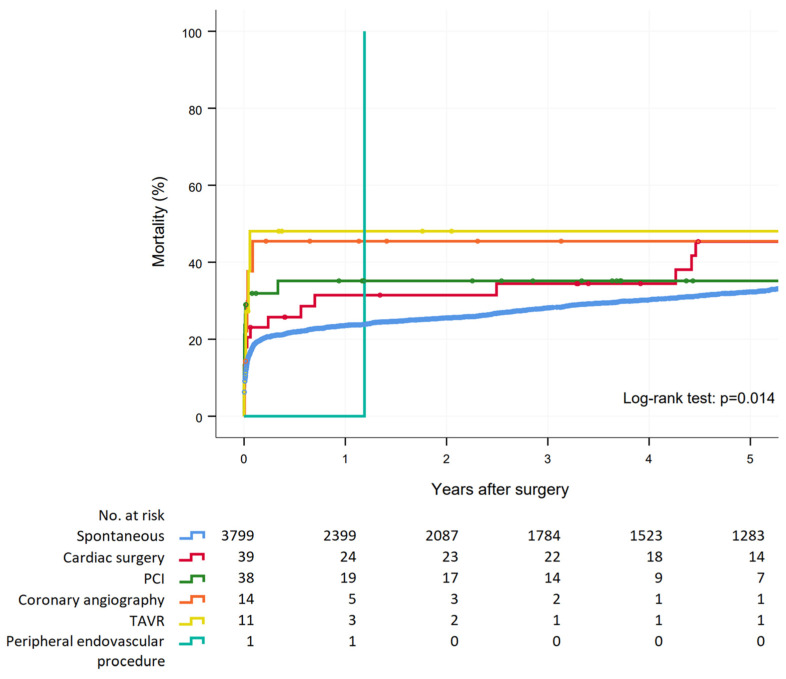
Survival estimates after surgery for acute type A spontaneous aortic dissection and iatrogenic aortic dissection secondary to cardiac surgery, percutaneous coronary intervention (PCI), coronary angiography, peripheral vascular intervention and transcatheter aortic valve replacement (TAVR).

**Table 1 jcm-11-06729-t001:** Characteristics and operative data of patients with spontaneous and iatrogenic acute TAAD in the unmatched and propensity score matched cohorts.

	Unmatched Cohorts			Propensity Score Matched Cohorts		
Characteristics	Spontaneous TAAD No. 3799	Iatrogenic TAAD No. 103	Standardized Differences	Spontaneous TAAD No. 102	Iatrogenic TAAD No. 102	Standardized Differences
Age, y	63.2 (13.1)	69.2 (9.4)	0.528	69.8 (10.9)	69.2 (9.4)	0.066
Females	1138 (30.0)	47 (45.6)	0.328	53 (52.0)	46 (45.1)	0.137
Bicuspid aortic valve	147 (3.9)	4 (3.9)	0.001	3 (2.9)	4 (3.9)	0.054
Diabetes	186 (4.9)	10 (9.7)	0.186	11 (10.8)	10 (9.8)	0.032
Stroke	147 (3.9)	6 (5.8)	0.091	4 (3.9)	6 (5.9)	0.091
Pulmonary disease	313 (8.2)	14 (13.6)	0.172	13 (12.7)	14 (13.7)	0.029
Extracardiac arteriopathy	181 (4.8)	18 (17.5)	0.413	14 (13.7)	18 (17.6)	0.108
Prior cardiac surgery	87 (2.3)	1 (1.0)	0.104	0 (0)	1 (1.0)	0.141
Severe aortic regurgitaion	706 (18.7)	9 (8.8)	0.364	16 (15.7)	9 (8.8)	0.210
Cardiac massage	159 (4.2)	8 (7.8)	0.151	6 (5.9)	8 (7.8)	0.078
Cardiogenic shock on inotropes	631 (16.6)	17 (16.5)	0.003	17 (16.7)	17 (16.7)	0.000
Cerebral malperfusion	819 (21.6)	10 (9.7)	0.331	9 (8.8)	10 (9.8)	0.034
Spinal malperfusion	80 (2.1)	2 (1.9)	0.012	1 (1.0)	2 (2.0)	0.082
Renal malperfusion	358 (9.4)	6 (5.8)	0.136	7 (6.9)	6 (5.9)	0.040
Mesenteric malperfusion	159 (4.2)	3 (2.9)	0.069	2 (2.0)	3 (2.9)	0.063
Peripheral malperfusion	535 (14.1)	8 (7.8)	0.204	7 (6.9)	8 (7.8)	0.038
DeBakey type I dissection	3205 (84.7)	32 (68.6)	0.399	66 (64.7)	70 (68.6)	0.083
Salvage procedure	221 (5.8)	7 (6.8)	0.040	4 (3.9)	5 (4.9)	0.048
Aortic arch replacement	771 (20.3)	5 (4.9)	0.479	16 (15.7)	15 (14.7)	0.027
Aortic root replacement	1082 (28.5)	15 (14.6)	0.344	6 (5.9)	7 (6.9)	0.040
CABG	306 (8.1)	46 (44.7)	0.913	3 (2.9)	46 (45.1)	1.000

Values are number (percentages) or mean (standard deviation). CABG = coronary artery bypass grafting; TAAD = type A aortic dissection.

**Table 2 jcm-11-06729-t002:** Characteristics and operative data of patients with iatrogenic acute TAAD according to the main causes of aortic dissection.

Characteristics	Cardiac Surgery No. 39	PCI No. 38	Coronary Angiography No. 38	TAVR No. 11	Peripheral Endovascular Procedure No. 1	*p*-Value
Delay from symptoms to surgery, h	6.3 (16.0)	13.7 (19.3)	17.8 (29.2)	3.4 (1.8)	36.0 (0)	<0.0001
Age, y	72.0 (6.3)	63.9 (9.3)	67.9 (10.1)	77.9 (8.0)	69.6 (0)	<0.0001
Females	15 (38.5)	19 (50.0)	5 (35.7)	7 (63.6)	1 (100	0.369
Bicuspid aortic valve	1 (2.6)	0 (0)	1 (7.1)	2 (18.2)	0 (0)	0.085
Diabetes	5 (12.8)	3 (7.9)	1 (7.1)	1 (9.1)	0 (0)	0.940
Stroke	2 (5.1)	4 (10.5)	0 (0)	0 (0)	0 (0)	0.529
Pulmonary disease	8 (20.5)	3 (7.9)	0 (0)	3 (27.3)	0 (0)	0.150
Extracardiac arteriopathy	10 (25.6)	5 (13.2)	2 (14.3)	1 (9.1)	0 (0)	0.535
Prior cardiac surgery	0 (0)	0 (0)	1 (7.1)	0 (0)	0 (0)	0.170
Cardiac massage	2 (5.1)	4 (10.5)	1 (7.1)	1 (9.1)	0 (0)	0.924
Cardiogenic shock on inotropes	5 (12.8)	7 (18.4)	2 (14.3)	3 (27.3)	0 (0)	0.798
Cerebral malperfusion	2 (5.1)	4 (10.5)	3 (21.4)	1 (100)	0 (0)	0.009
Spinal malperfusion	1 (2.6)	1 (2.6)	0 (0)	0 (0)	0 (0)	0.953
Renal malperfusion	3 (7.7)	3 (7.9)	0 (0)	0 (0)	0 (0)	0.708
Mesenteric malperfusion	2 (5.1)	1 (2.6)	0 (0)	0 (0)	0 (0)	0.832
Peripheral malperfusion	5 (12.8)	2 (5.1)	1 (7.1)	0 (0)	0 (0)	0.602
DeBakey type I dissection	23 (59.0)	29 (76.3)	11 (78.6)	6 (54.5)	1 (100)	0.649
Salvage procedure	2 (5.1)	3 (7.9)	1 (7.1)	1 (9.1)	0 (0)	0.982
Aortic arch replacement	1 (2.6)	2 (5.3)	1 (7.1)	0 (0)	0 (0)	<0.0001
Aortic root replacement	5 (12.8)	3 (7.9)	4 (28.6)	3 (27.3)	0 (0)	0.262
CABG	17 (43.6)	24 (63.2)	5 (35.7)	0 (0)	0 (0)	0.004

Values are number (percentages) or mean (standard deviation). CABG = coronary artery bypass grafting; PCI = percutaneous coronary intervention; TAAD = type A aortic dissection; TAVR = transcatheter aortic valve replacement.

**Table 3 jcm-11-06729-t003:** Cardiovascular procedures resulting in iatrogenic acute TAAD and their related in-hospital and 5-year mortality rates.

Procedures	No. (%)	In-Hospital Mortality	5-Year Mortality
Cardiac surgery	39 (37.8)	8 (20.5)	16 (47.5)
Isolated CABG	17 (16.5)		
Mitral valve surgery	8 (7.8)		
Mitral valve surgery, CABG	3 (2.9)		
AVR	5 (4.9)		
AVR, ascending aortic aneurysm replacement	1 (9.7)		
Ascending aortic aneurysm replacement	2 (1.9)		
AVR, tricuspid valve repair	2 (1.9)		
David procedure	1 (9.7)		
Percutaneous coronary intervention	38 (36.9)	12 (31.6)	13 (35.2)
Coronary angiography	14 (13.6)	6 (42.9)	6 (45.5)
Transcatheter aortic valve replacement	11 (10.7)	5 (45.5)	5 (48.1)
Intracranial endovascular procedure	1 (1.0)	0 (0)	1 (100)

Values are numbers (percentages). AVR = aortic valve replacement; CABG = coronary artery bypass grafting; TAAD = type A aortic dissection.

**Table 4 jcm-11-06729-t004:** Early and late outcomes of patients with spontaneous and iatrogenic acute TAAD in the unmatched and propensity score matched cohorts.

	Unmatched Cohorts			Propensity Score Matched Pairs		
Outcomes	Spontaneous TAAD No. 3799	Iatrogenic TAAD No. 103	*p*-Value	Spontaneous TAAD No. 102	Iatrogenic TAAD No. 102	*p*-Value
Early outcomes						
In-hospital mortality	658 (17.3)	31 (30.1)	0.001	16 (15.7)	31 (30.4)	0.013
Neurological complications	812 (21.4)	22 (21.4)	0.997	20 (19.6)	22 (21.6)	0.729
Stroke	575 (15.1)	17 (16.5)	0.702	15 (14.7)	17 (16.7)	0.700
Ischemic stroke	525 (13.8)	14 (13.6)	0.947	15 (14.7)	14 (13.7)	0.841
Hemorrhagic stroke	73 (1.9)	3 (2.9)	0.453	0 (0)	3 (2.9)	0.246
Global brain ischemia	175 (4.6)	2 (1.2)	0.330	3 (2.9)	2 (2.0)	1.000
Paraplegia/paraparesis	198 (5.2)	6 (5.8)	0.783	3 (2.9)	6 (5.9)	0.498
Tetraplegia	3 (0.1)	0 (0)	1.000	0 (0)	0 (0)	-
Composite end-point ^a^	1107 (29.1)	45 (43.7)	0.001	29 (28.4)	45 (44.1)	0.020
Heart failure	516 (13.6)	36 (35.0)	<0.0001	11 (10.8)	36 (35.3)	<0.0001
Mechanical circulatory support	128 (3.4)	13 (12.6)	<0.0001	2 (2.0)	13 (12.7)	0.005
IABP	29 (0.)	6 (5.8)	<0.0001	1 (1.0)	6 (5.9)	0.119
VA-ECMO	102 (2.7)	9 (8.7)	0.002	1 (1.0)	9 (8.8)	0.019
Dialysis	543 (14.3)	16 (15.5)	0.725	13 (12.7)	16 (15.7)	0.548
Sepsis	464 (12.2)	10 (9.7)	0.443	13 (12.7)	10 (9.8)	0.507
Laryngeal nerve palsy	71 (1.9)	0 (0)	0.264	0 (0)	0 (0)	-
Reoperation for intrathoracic bleeding	530 (14.0)	19 (18.4)	0.195	15 (14.7)	19 (18.6)	0.452
Tracheostomy	305 (8.0)	14 (13.6)	0.042	6 (5.9)	14 (13.7)	0.060
Deep sternal wound infection	84 (2.2)	5 (4.9)	0.085	2 (2.0)	5 (4.9)	0.445
Mesenteric ischemia	145 (3.8)	4 (3.9)	0.973	3 (2.9)	4 (3.9)	1.000
Acute lower limb ischemia	120 (3.2)	4 (3.9)	0.569			
Acute upper limb ischemia	12 (0.3)	1 (1.0)	0.294	0 (0)	4 (3.9)	0.121
Additional procedures						
Revasc. for mesenteric ischemia	7 (0.2)	0 (0)	1.000	0 (0)	0 (0)	-
Revasc. for renal ischemia	7 (0.2)	0 (0)	1.000	0 (0)	0 (0)	-
Revasc. for lower limb ischemia	51 (1.3)	1 (1.0)	1.000	0 (0)	1 (1.0)	1.000
Revasc. for upper limb ischemia	7 (0.2)	1 (1.0)	0.193	0 (0)	0 (0)	-
Major lower limb amputation	17 (0.4)	0 (0)	1.000	0 (0)	0 (0)	-
Surgery for intestinal complications	18 (0.5)	0 (0)	1.000	0 (0)	0 (0)	-
Late outcome						
5-year mortality	1087 (32.3)	41 (45.6)	0.002	34 (39.4)	41 (46.2)	0.163

Values are number (percentages). Revasc = revascularization. IABP = intra-aortic balloon pump; TAAD = type A aortic dissection; VA-ECMO = venoarterial extracorporeal membrane oxygenation. ^a^ Includes in-hospital death, stroke and/or global ischemia.

## Data Availability

Data from this registry is not publicly available.

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
