# Peer review of "Outcome after Surgery for Iatrogenic Acute Type A Aortic Dissection"

_jcm, 2022, doi:10.3390/jcm11226729_

Round 1

Reviewer 1 Report

This is an original article, based on an international registry on Aortic Dissection, and reports results regarding iatrogenic TAAD. Aim, methods and results are very clearly presented and are of high interest for readers regardless their specialty. Aortic dissection remains a deadly surgical pathology and iTAAD is feared by both surgeons and interventional cardiologists. The results of the reviewed article are therefore very interesting. English language is very correct. I would reccommend acceptance in its current form. 

Author Response

1. This is an original article, based on an international registry on Aortic Dissection, and reports results regarding iatrogenic TAAD. Aim, methods and results are very clearly presented and are of high interest for readers regardless their specialty. Aortic dissection remains a deadly surgical pathology and iTAAD is feared by both surgeons and interventional cardiologists. The results of the reviewed article are therefore very interesting. English language is very correct. I would reccommend acceptance in its current form.

Response: We are grateful to the Reviewer for her/his kind comments.

Changes: None

Reviewer 2 Report

This study sought to investigate the outcome after surgery for Iatrogenic acute type A aortic dissection (TAAD). The primary outcomes (in-hospital mortality and 5-year mortality) and 27 secondary outcomes were evaluated in data from the European Registry of Type A Aortic Dissection (ERTAAD) study. They revealed that Iatrogenic origin of acute TAAD is quite uncommon but carries a significantly increased risk of in-hospital mortality compared to spontaneous TAAD.

Major comments:

1.    This article lacks innovation. Data related to Iatrogenic TAAD have been reported several times, e.g., references in this paper (1, 3-11). The present study did not find significantly different conclusions from previous studies.

2.    The cases included in the statistics of this study span 16 years (from January 1, 2005 to March 31, 2021). Over the past 16 years, various surgical techniques have advanced and improved. Therefore, does the occurrence of iatrogenic TAAD due to surgery change over time? I suggest that the authors compare the changes in earlier and more recent relevant data.

Author Response

1. This article lacks innovation. Data related to Iatrogenic TAAD have been reported several times, e.g., references in this paper (1, 3-11). The present study did not find significantly different conclusions from previous studies.

Response: We are grateful to the Reviewer for these comments. We respectfully disagree with her/his opinion, because the findings of this study are different from those of previous studies. In fact, previous studies reported similar results compared to spontaneous TAAD. Furthermore, no analysis of the results between different causes of iatrogenic TAAD has been ever performed. This is the first study including patients who underwent surgery for TAAD after transcatheter procedures.

Changes: None.

2. The cases included in the statistics of this study span 16 years (from January 1, 2005 to March 31, 2021). Over the past 16 years, various surgical techniques have advanced and improved. Therefore, does the occurrence of iatrogenic TAAD due to surgery change over time? I suggest that the authors compare the changes in earlier and more recent relevant data.

Response: We thank the Reviewer for these thoughtful comments. We performed new analyses to evaluate any changes in the earlier and more recently operated patients.

Changes: We added the following sentences to the Results section: “We evaluated the results of earlier and more recent procedures by comparing the results of patients operated before 2014 (44 patients, 42.7%) and those operated later on (59 patients, 57.3%) adjusted for independent risk factors for in-hospital mortality along with types of procedures. Patients operated more recently did not have a lower risk of in-hospital mortality (p=0.161, OR 3.330, 95%CI 0.619-17.901)

Reviewer 3 Report

I think this paper is important for many readers.

The patients are large, and 5 years follow up is good.

My concerns are follows.

The number of the iatrogenic AD was small. Therefore the conclusion is not trustworthy.

The patients with PCI was 38. On the other hand, TAVR was only 11. It is questionable.

The representative cases are required.

The following situation of iatrogenic AD are required, e.g. tamponade, AR…

The duration for diagnosis are needed.

Round 2

Reviewer 2 Report

I have no other comment.